# Probabilistic Model for Aero-Engines Fleet Condition Monitoring

**Valentina Zaccaria [1,\*], Amare D. Fentaye [1], Mikael Stenfelt [1,2] and Konstantinos G. Kyprianidis [1]**

[1]  Simulation and Optimization for Future Industrial Applications, Mälardalen University,
    72123 Västerås, Sweden; amare.desalegn.fentaye@mdh.se (A.D.F.); mikael.stenfelt@mdh.se (M.S.);
    konstantinos.kyprianidis@mdh.se (K.G.K.)
[2]  SAAB Aeronautic, 58254 Linköping, Sweden
\*  Correspondence: valentina.zaccaria@mdh.se

**Abstract:** Since aeronautic transportation is responsible for a rising share of polluting emissions, it is of primary importance to minimize the fuel consumption any time during operations. From this perspective, continuous monitoring of engine performance is essential to implement proper corrective actions and avoid excessive fuel consumption due to engine deterioration. This requires, however, automated systems for diagnostics and decision support, which should be able to handle large amounts of data and ensure reliability in all the multiple conditions the engines of a fleet can be found in. In particular, the proposed solution should be robust to engine-to-engine deviations and different sensors availability scenarios. In this paper, a probabilistic Bayesian network for fault detection and identification is applied to a fleet of engines, simulated by an adaptive performance model. The combination of the performance model and the Bayesian network is also studied and compared to the probabilistic model only. The benefit in the suggested hybrid approach is identified as up to 50% higher accuracy. Sensors unavailability due to manufacturing constraints or sensor faults reduce the accuracy of the physics-based method, whereas the Bayesian model is less affected.

**Keywords:** diagnostics; performance model; Bayesian network; turbofan; fleet

## 1. Introduction

The stringent goals of reduced $CO_2$ emissions set by the International Civil Aviation Organization (ICAO) represent a challenge for aero-engine manufacturers, who are pushing the design to extreme conditions and incrementing the use of alternative fuels [1]. With increasing design complexity and more challenging operating conditions, airlines must rely on effective systems for monitoring, diagnostics, and prognostics. Accurate diagnosis and prognosis of engine health condition is surely necessary to reduce maintenance cost, since engines are currently accounting for about 25% of the operating cost of an aircraft [2]. Furthermore, it is also essential to optimize flight operations depending on the health status of the engine and avoid unnecessary efficiency decrement and consequent excessive fuel consumption.

Airlines have to deal with large fleets of engines. Advances in acquisition systems and computational tools have rendered available a huge amount of data for diagnostics: high-resolution, on-wing data are now often available instead of a single flight data point. However, this demands fully automated diagnostic systems that can effectively replace human reasoning. The challenges that such a diagnostic system must face are multiple. Besides the capability of processing large amounts of data in a reasonable time, robustness toward engine-to-engine variations due to serial deviations and different degradation history is a pressing demand.

Adaptive and regression models to mitigate the effects of engine-to-engine variations have been proposed in the literature [3–6]. Chu et al. developed scalable regression models for a fleet of aircraft engines for anomaly detection to solve two major issues: performance variability within the fleet due to production scatter and potential computational burden due to the large amount of collected data [3,4]. The algorithm was tested on three scenarios: performance anomaly detection, performance shift detection over different flights, and a malfunctioning aircraft, implanting six cases over 200 aircraft. A scalable approach for anomaly detection of a large population of aircraft was also proposed by Ohlsson et al., with the important advantage of no training needed for the algorithm [6].

Diagnostics of a gas turbine comprises of three fundamental steps: first, anomaly detection, in which a malfunction or an anomalous condition is detected based on measured data; second, fault isolation, which determines the component(s) where the possible malfunction is located; and third, fault identification, which determines the fault severity and hence provides precious information for possible corrective actions [7]. The numerous techniques for fault detection, isolation, and identification can be generally grouped into physics-based methods [8,9] (mostly based on the gas path analysis proposed by Urban [10]), data-driven methods [11], and hybrid methods [12]. Extensive reviews of gas turbine diagnostic methods for aircraft engines and land-based turbomachines exist in the literature [7,13]. The most common faults that affect aero-engine performance over the lifetime are compressor fouling, which is usually characterized by a decrease in both compressor efficiency and flow capacity, and turbine erosion, which often leads to decreased efficiency and increased flow capacity.

Gas path analysis (GPA) based on performance models of the engine is still one of the most common techniques. Various non-linear and adaptive models [14,15], also in combination with machine learning methods [16,17], have been proposed to overcome the typical shortcomings of GPA, such as smearing effect and high sensitivity to measurement noise [18,19]. Many efforts over the years have been made on identifying optimal sets of measurements and optimal combinations of measured values and performance factors to achieve desired diagnostics capability [9,20]. A limitation in the use of performance models is the frequent lack of proprietary information about commercial units, in case the service provider is not the manufacturer. A way around this limitation and retaining high diagnostics accuracy has been achieved by employing machine learning techniques such as neural network [21], support vector machine [22], Bayesian network [23], etc. A common challenge when applying any data-driven method to a fleet of engines is the large amount of data needed for training, and the limited robustness toward serial deviations, as previously discussed. Rarely the proposed solutions in the open literature are tested on a (real or simulated) fleet.

Due to its intrinsic capability to deal with uncertain information, a Bayesian network (BN) has been used very often for constructing a probabilistic model for diagnostics purpose. The advantage of BN over other data-driven methods is that expert knowledge can be combined with information from data, reducing the need for historical data and relying more on the engineering knowledge. The applications on fault classification for gas turbines have demonstrated the success of this method [23–26]. Sensor validation through hierarchical BN models was also proposed by different authors [27,28]. One additional application explored in recent decades is the use of BN for information fusion; for instance, Kyriazis et al. fed the outcomes of various diagnostics methods to a BN to establish the most likely fault scenario [29,30]. All these examples were proven successful when applied to a single engine, but no study on the effect of engines serial deviation or operating conditions was performed.

Among the work done on fleet monitoring, Martinez et al. proposed a fuzzy logic-based technique for engine health monitoring (EHM) that classified the engines of the fleet in four classes: good, normal, highly deteriorated, and bad [31]. Their approach was a significant step toward a method to minimize maintenance cost in a fleet. A GPA approach was presented and tested on flight data from a KLM aircraft assessing the effect of sensor noise, but complete fleet monitoring and diagnostics were not presented [32]. Kraft et al. proposed a method to optimize the maintenance schedule in a fleet based on a novel high-fidelity modeling approach [33]. The authors claimed that a reduction of 10% in maintenance cost was achieved with their solution.

In this work, we are presenting the development of a tool for gas turbine diagnostics that combines an adaptive performance model and a probabilistic BN for fault detection and identification. The advantages of both strategies are exploited by a combination of the two:

- Model errors and uncertainties are reduced in the final outcome because of model adaptation to baseline values;
- Smearing effects are mitigated in the BN layer;
- Data correction is provided via model adaptation, improving the BN performance;
- Soft sensors from the model can be included in the BN to increase the accuracy.

The robustness and flexibility of the integrated hybrid system are demonstrated by testing it on a fleet of engines characterized by serial deviations, different deterioration levels, and various flight conditions. The novelty of the work is to compare benefits and shortcomings of combined physics-based and data-driven methods when applied to the whole fleet rather than a single engine.

## 2. Diagnostics Framework Description

The diagnostic framework used in this work consists of a multi-layer approach. First, the unreliable measurements such as values out of physical boundaries (e.g., a negative pressure) or frozen measurements are discarded, and subsequently, the remaining measurements are fed to a physics-based adaptive model for a first fault detection and isolation procedure. Finally, a probabilistic BN is used in combination with the performance model for fault isolation and identification. In this work, the benefits of the proposed solution for fleet monitoring and diagnostics will be demonstrated.

The purpose of the adaptive model is twofold. During the first performance test, the measured data from the healthy engine are compared to the model outputs and the model performance parameters vary to minimize the error between the two; the obtained adapted model at baseline conditions (denoted as *AD,ref* in the following sections) includes serial deviations from the nominal engine. This adaptive baseline can be used to calculate measurement deviations (or residuals) during the engine lifetime. The second purpose is to correct the measurements with respect to ambient conditions and flight conditions, as further explained in the following sub-sections.

In this work, three methods are compared: the use of the BN method only, the use of the adaptive model for gas path analysis, and the hybrid method that combines both.

### 2.1. The Adaptive Model

The model under consideration simulates the performance of a three-shaft turbofan engine. The model was developed with the in-house software EVA and previously presented [34]. The various components of the engine, depicted in Figure 1, were modeled according to mass and energy balance equations, Gibbs free energy equations, and representative components maps, under the assumption of ideal gases. The adaptation scheme was based on the minimization of residuals between target values and model outputs, which form a square Jacobian matrix.

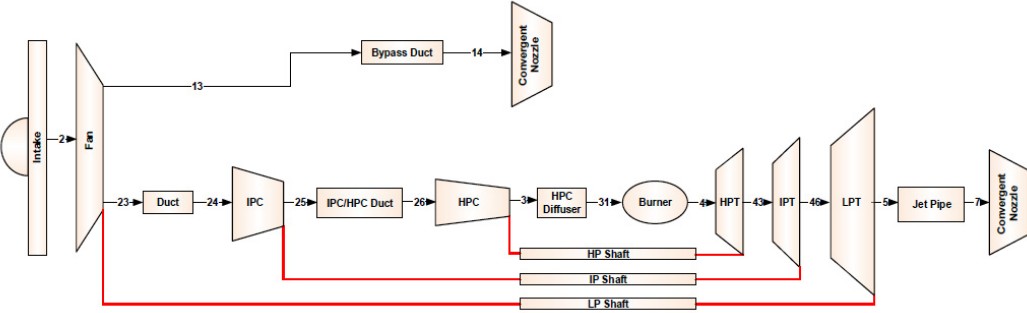

**Figure 1.** Schematic diagram of the modeled engine.

The rows of the matrix represent the residuals between model outputs and measurements while the columns represent state variables that are varied at each iteration until all the residuals are below a predefined threshold. The Jacobian is a square matrix, i.e., the number of state variables and target values has to be the same. Each state variable is automatically associated with a residual depending on the impact that the state variable has on the corresponding output. For diagnostic purposes, the state variables that the model has to adjust include the performance deviation factors, i.e., the deviation factor in isentropic efficiency and flow capacity for each component:

- $\Delta\eta_{IPC}$, $\Delta\bar{W}_{IPC}$
- $\Delta\eta_{HPC}$, $\Delta\bar{W}_{HPC}$
- $\Delta\eta_{HPT}$, $\Delta\bar{W}_{HPT}$
- $\Delta\eta_{IPT}$, $\Delta\bar{W}_{IPT}$
- $\Delta\eta_{LPT}$, $\Delta\bar{W}_{LPT}$

Please note that in this work, only the intermediate and high pressure compressors and the three turbines were considered to illustrate the proposed method. Deviation factors are defined as:

$$\Delta\eta = \eta_{fault} - \eta_{design} \tag{1}$$

$$\Delta\bar{W} = \frac{\bar{W}_{fault}}{\bar{W}_{design}} \tag{2}$$

The target values are instead coming from the sensor measurements; it is hence notable in this approach that the number of available measurements needs to match the number of considered performance deviation factors. If this is not the case, relationships between components performance deviations have to be assumed to reduce the number of state variables (for example the ratio between efficiency deterioration in the three turbines could be fixed, or the ratio between efficiency and flow capacity deterioration in each component). The effect of varying the number of sensors was evaluated in this work.

The selection of sensors in an ideal case can be based on the following criteria:

- Eliminating measurements with high mutual correlation (>90%), since no additional useful information is provided by a measurement that is highly correlated with an already existing one;
- Selection of the measured values that are mostly affected by a change in the considered state variables.

The model can run ten thousands of data points in a few seconds on a normal computer, making it a suitable tool for monitoring and diagnostics of an entire fleet.

### 2.2. The Matching Scheme

Typically, a diagnostics problem consists of finding a state vector $x$ that contains all the performance deviations (e.g., $\Delta\eta$ and $\Delta\bar{W}$ for each component) from the measurement deviations vector $z$, where all the measurements are normalized with the value at reference condition, as illustrated in Figure 2. As shown in [35], we can define an influence matrix $H$ as in Equation (3).

$$z = Hx \tag{3}$$

There are two aspects in measurement selection, one purely mathematical concerning which variables would be theoretically more suitable for identification of specific faults [36,37], and the other practical concerning which sensors are actually available on an aircraft. Measurements selection was first performed on a theoretical basis to evaluate faults observability; in this step, all the output variables were considered even if normally not measured in a real engine. Each deviation parameter was varied by 1% one at a time and the resulting influence matrix was analyzed, which contains deviations in measured values caused by a unitary variation in state variable, as illustrated in Figure 2.

| | $\dfrac{N_{46} - N_{46,ref}}{N_{46,ref}}$ | $\dfrac{T_{25} - T_{25,ref}}{T_{25,ref}}$ | $\dfrac{p_{25} - p_{25,ref}}{p_{25,ref}}$ | $\dfrac{T_3 - T_{3,ref}}{T_{3,ref}}$ | ... |
|---|---|---|---|---|---|
| Δη_IPC | -0.4 | 0.17 | -0.6 | 0.3 | |
| Δw_IPC | -0.3 | 0.16 | -0.3 | 0.15 | |
| Δη_HPC | -0.2 | 0.09 | 0.8 | 0.11 | |
| Δw_HPC | -0.004 | 0.02 | 0.2 | 0.05 | |
| Δη_HPT | -0.2 | 0.1 | 1.04 | -0.34 | |
| ... | | | | | |

**Figure 2.** Illustrative example of influence matrix *H*.

Correlation between measurement vectors was calculated (i.e., between the matrix columns) as in [38]. Correlation between columns *i* and *j* of the influence matrix is expressed in Equation (4), where M is the length of the measurement vectors $z_i$ and $z_j$.

$$c_{ij} = \frac{z_i \cdot z_j}{\sqrt{\sum_{m=1}^{M} z_{i,m}^2 \sum_{m=1}^{M} z_{j,m}^2}} \tag{4}$$

A high correlation level was considered if larger than 90%; the measurements with a higher number of high correlation with other sensor measurements were discarded, until no correlated measurements remained. Successively, the influence of each performance deviation factor on the remaining measurements was analyzed, i.e., on each element of the influence matrix in the sensor's column. Measurements were discarded if a deviation smaller than three times the sensor standard deviations was observed for all simulated faulty cases, because the induced deviations were comparable to measurement noise. In other words, for a simulated fault case to be included, its induced deviations must be larger than sensors uncertainty.

Finally, the correlation between matrix rows was calculated (i.e., between measurements deviations induced by different faults) also with Equation (4), which showed a quite large correlation between Δη_HPC and Δη_HPT (89%). This suggests that correct identification of these two deviation parameters may be challenging in this engine configuration. The sensors selection results for this engine are shown in Section 4.

The adaptation scheme was also successfully applied in previous work to a 50 MW industrial gas turbine [39] and a micro gas turbine for heat and power generation [40], demonstrating the flexibility of the approach. This represents the first use of the adaptive model.

The second use of the adaptive model, as presented in [39], is for data correction with respect to ambient and flight conditions, which allows combining this physics-based approach with a data-driven diagnostic method for fault classification and identification. As such, the adaptation scheme can run in each flight condition, since the values of ambient temperature and pressure are set as inputs in the model, from which altitude and Mach number can be calculated. Once the performance deviation factors $(\Delta\eta, \Delta\overline{W})_{det}$ are estimated by the adapted model at flight condition, *AD,flight**, those factors are used in the model to simulate the same engine at reference conditions (nominal flight condition at ISA temperature), which is denoted as *AD,ref**. With this approach, the effect of varying ambient temperature and flight parameters such as altitude or Mach number is isolated and removed from the detected performance deviations. The flowchart of the algorithm is shown in Figure 3, where solid arrows represent data from measurements and dashed arrows represent the estimated deviation factors in efficiency and flow capacity.

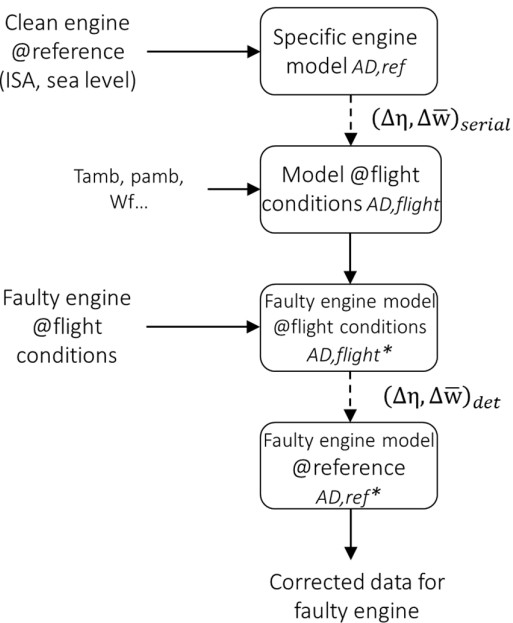

**Figure 3.** Flowchart of fault detection and data correction procedure with the adaptive model.

*2.3. Bayesian Network Fault Classification*

Once the measured data are corrected with respect to ambient and flight conditions through the model at reference conditions, their residuals with respect to healthy reference conditions can be used in a data-driven algorithm to identify the cause of the engine performance deviation. The advantage of coupling the adaptive model with a data-driven algorithm is that other malfunctions such as mechanical failures (e.g., a valve leakage) or sensor failures could also be detected. In fact, numerous mechanical faults show up as deviations in thermodynamic parameters, and a further analysis is needed to isolate the correct cause. In addition, more measurements could be included in this second step e.g., vibration measurements but also soft sensors (i.e., model output variables that are not measured), which enhances the capability of the diagnostic system.

A Bayesian network is a graphical probabilistic model that represents the probabilistic relationships between causes and effects according to the Bayes theorem, reported in Equation (5). The equation represents a casual statement of the kind $X \rightarrow Y$, where $X$ causes $Y$ and $Y$ takes the role of an observable effect of $X$. $P(Y)$ is called the prior probability, while $P(Y|X)$ is called the posterior probability. The factor that relates the two, $P(X|Y)/P(X)$, is called the likelihood ratio.

$$P(Y|X) = \frac{P(X|Y)}{P(X)} P(Y) \tag{5}$$

A BN comprises of a direct acyclic graph (DAG) that includes nodes and edges, and a conditional probability table (CPT) for each node. The DAG is the graphical representation of the model, where the causes are represented with parent nodes and the effects with child nodes; the edges represent conditional dependencies and connect causes and effects in an acyclic way, i.e., it is impossible to start from a node and traverse the entire network following edges and subsequent nodes.

Both DAG structure and CPTs can be either constructed by experience or learned from data. Hybrid methods are also possible, where the human experience intervenes for incomplete or incorrect data. In this work, the BN structure was built manually, while the CPTs were fitted from data generated by the performance model. The selected BN model included five parent nodes, one for each component (IPC, HPC, HPT, IPT, and LPT), and one child node for each sensor measurement. Parent nodes were defined with four possible states, corresponding to different deterioration levels: Normal (N) for no performance deviation, Very Low (VL) for performance deviations within

manufacturing tolerance, Low (L) for fault severity between 1.1% and 2.2%, and Medium (M) for fault severity between 2.2% and 3.3%. The fault severity $S$ was a function of efficiency and flow capacity deviation calculated according to Simon [41] as in Equation (6). This work focused on small fault severity magnitude because they are usually more difficult to detect. Fault severities higher than 3% could also be included by increasing the number of states in the parent nodes.

$$S = -\Delta\eta \cdot \sqrt{1 + \left(\frac{\Delta\overline{w} - 1}{\Delta\eta}\right)^2} \tag{6}$$

The ratio between the flow capacity deviation percentage $\Delta\overline{W} - 1$ and the efficiency deviation was assumed constant and equal to 2 [41]. Each of the child nodes was instead defined with 20 states. These states represent non-overlapping intervals of measurements deviations from the reference value, and were selected by dividing the span between minimum and maximum deviations into 20 intervals. The network was built in the HUGIN Expert software environment [42] and the Expectation Maximization algorithm was applied to estimate the CPTs from training data.

An additional parent node was added for subsequent tests representing a bleed valve (BV) leakage, with three states: N (0% leak), L (up to 2% leak), and M (between 2% and 4% leak). These tests will illustrate how additional faults such as valves leakage or mechanical failures can be included in the BN classification and in the hybrid approach.

The BN inference can of course be applied directly to measured data instead of adapted data from the performance model; this scenario was compared to the proposed strategy to assess the benefit of a hybrid method over a purely data-driven one. When measured data were used directly for the data-driven fault classification, data correction with ambient conditions was performed as suggested by Volponi [43]. Measured parameters were corrected with temperature and pressure ratios ($\theta = T_{amb}/T_{ref}$, $\Delta = p_{amb}/p_{ref}$) according to Equation (7), where the exponents $a$ and $b$ were set as suggested in [43].

$$z_{corr} = \frac{z}{\theta^a \delta^b} \tag{7}$$

## 3. Methodology

The objective of the work was to test the diagnostic framework on a fleet of engines to demonstrate the robustness of the approach. Different units of the same engine differ because of manufacturing tolerances, which make them perform differently even when brand new. During operations, the differences are enlarged because of different flight conditions experienced. Generally, data-driven techniques have to be trained on a space that considers all possible serial deviations, measurements noise, and flight conditions, and their performance reduces if the engine conditions change significantly. These reasons make the use of such techniques for diagnostics challenging.

Fleet data were generated by applying a Gaussian distribution to each performance deviation factor as shown in Table 1 and simulating in this way 200 engines that differ in design efficiency and flow capacity for each rotating component. Standard deviation values were chosen consistently with common engine performance as per the Author's experience. It is important to note that a deviation factor on the isentropic efficiency was added to the reference value ($\eta_{ref} + \Delta\eta$), while a deviation factor in flow capacity was a coefficient multiplying the desired mass flow ($\overline{W}_{ref} * \Delta\overline{W}$), hence, the nominal value is 1.

**Table 1.** Gaussian distribution characteristics for engine serial deviations simulations (Case 1).

| Parameter | Mean | $3\sigma$ |
|-----------|------|-----------|
| $\Delta\eta$ | 0.00 | 0.5% |
| $\Delta\overline{W}$ | 1.00 | 1% |

To simulate variations in flight conditions, the deviations in efficiency and flow capacity were kept constant and some flight conditions parameters were varied with a uniform distribution as illustrated in Table 2. It is important to note that altitude and Mach number were set as model inputs to simulate the faulty engines, while for model adaptation, these parameters were calculated from static and total pressure measurements and hence subject to uncertainty.

**Table 2.** Uniform distributions characteristics for flight parameters (Case 2).

| Parameter | Max Value | Min Value |
|---|---|---|
| Altitude | 11,000 m | 10,000 m |
| Ambient temperature | $T_{ISA} + 15$ K | $T_{ISA} - 10$ K |
| Mach number | 0.85 | 0.82 |

The nominal values at Top of Climb (ToC) for the considered engine are listed in Table 3. The model is representative of a 311 kN engine with technology levels consistent with entry into service in 1995.

**Table 3.** Nominal conditions at ToC.

| Variable | Value |
|---|---|
| Flight altitude | 10,668 m |
| Flight Mach no. | 0.82 |
| Bypass ratio | 4.7 |
| Overall pressure ratio | 34 |
| HPT inlet temperature | 1625 K |
| Specific thrust | 167 N·s/kg |

For Case 1, half of the engines were used for training the BN (i.e., for estimating the CPTs) and half for test. For each of the 100 training engines, a deterioration pattern was simulated for each of the five components including 21 levels of fault severity ranging from 0% to 3.3%. As a result, 10,500 points formed the dataset for the BN. For Case 2, 100 random combinations of flight parameters were picked from the selected range and the same fault pattern (21 severity levels * 5 components) was simulated for each flight, for a total of other 10,500 data points.

For the tests, the 100 engines and the 100 flight parameters combinations that were not used for CPTs estimation were employed. For each engine and each flight condition, one healthy and three fault scenarios were simulated for each component, as presented in Table 4, for a total of 3200 tested points.

**Table 4.** Simulated fault scenarios.

| Fault Severity | $\Delta\eta$ | $\Delta\overline{W}$ |
|---|---|---|
| 0% (N) | 0% | 1.00 |
| 0.7% (VL) | −0.3% | 0.994 IPC/HPC<br>1.006 HPT/IPT/LPT |
| 1.8% (L) | −0.8% | 0.984 IPC/HPC<br>1.016 HPT/IPT/LPT |
| 2.9% (M) | −1.35% | 0.973 IPC/HPC<br>1.027 HPT/IPT/LPT |

Gaussian noise was added to all sensor data. As mentioned in Section 2, the results of the hybrid adaptive model and BN approach were compared with the results obtained by the BN alone and the outcomes of the adaptive model alone, following the scheme of Figure 4.

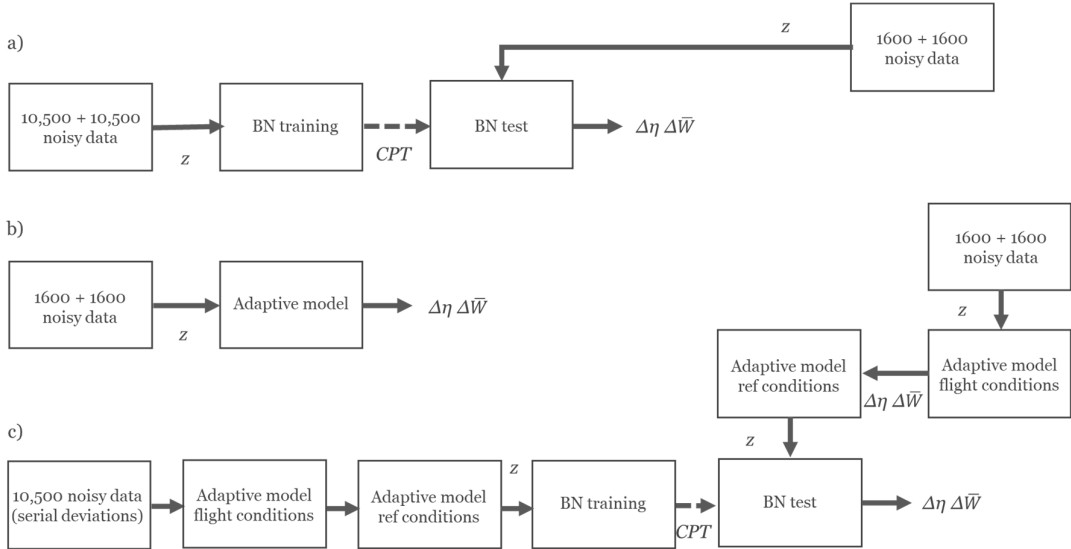

**Figure 4.** Schematic diagram of the three compared approaches: BN only (**a**), adaptive model only (**b**), and hybrid method with adaptive model and BN (**c**).

To test the capability to detect a mechanical fault, the training space was enlarged to include bleed valve leakage in each engine of the fleet and at each flight conditions ranging from 0 to 4%. Then, a BV leakage was simulated for the tested engines at 1.5% and 3% of the nominal flow.

## 4. Results

First, the most suitable sensor set for the model adaptation procedure was identified based on the method described in Section 2. This step was applied to determine which measurements would be required in an ideal case, and more realistic sensor sets were successively considered for comparison. The influence of each performance deviation factor on the uncorrelated measurements is shown in Figure 5 (for the variables number, refer to the diagram in Figure 1). The bars in the Figure represent the deviation in the normalized measurements as in the influence matrix of Figure 2 induced by 1% performance change. Where the induced deviation was smaller than $3\sigma$ (i.e., sensor standard deviation) for all simulated faulty cases, the sensors were discarded. For example, T43, T46, W2, and W13 were discarded at this stage. Measurements of fan and IPC inlet flow are often available during pass-off tests and it was hence interesting to evaluate their potential use in the adaptation scheme for baseline generation. However, their impact in the influence matrix led to discard them.

From the plots of Figure 5, it was noted that $\Delta \bar{W}_{HPC}$ had a very small influence on the available measurements; therefore, detection of HPC fault was expected to be more difficult. The sensors selected for the matching scheme are listed in Table 5 with considered noise level together with the sensors for the fuel flow and for ambient conditions.

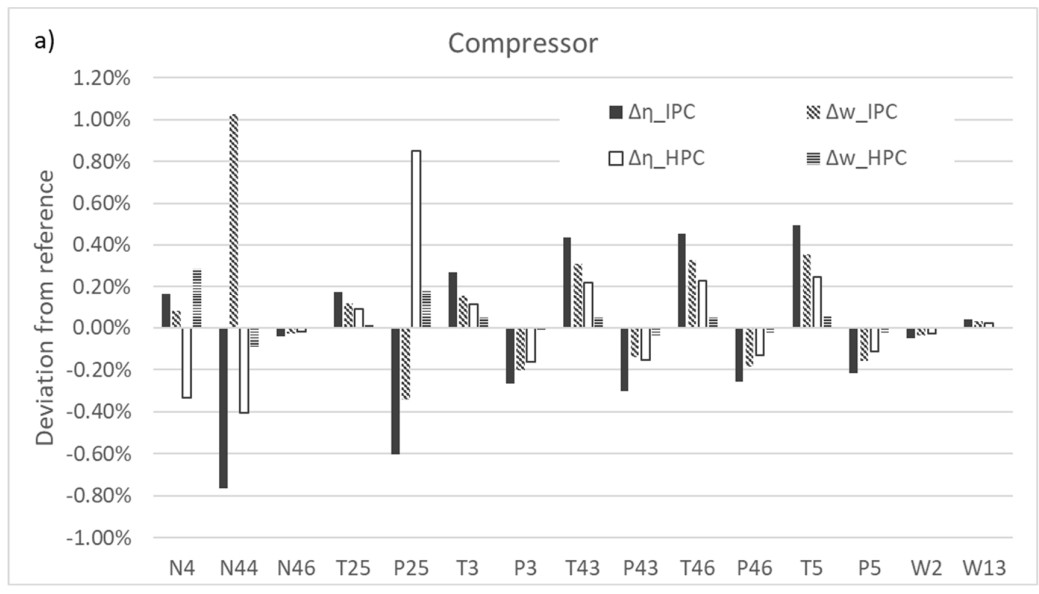

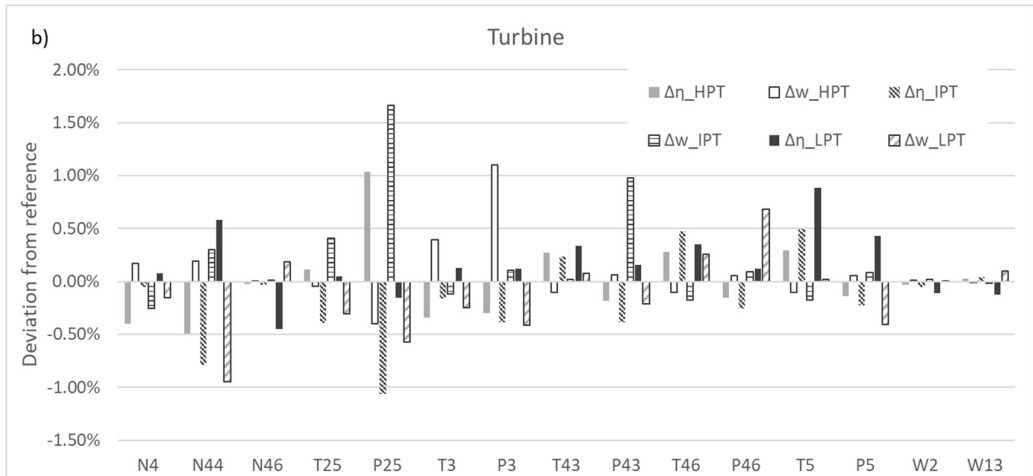

**Figure 5.** Influence of performance parameters deviation on measurements of compressors (**a**) and turbines (**b**).

**Table 5.** Sensors list with considered noise level.

| Sensor Name | Measured Variable | Noise (σ) |
|---|---|---|
| T1 | Ambient temperature | ±0.3% |
| p1 | Ambient pressure (static) | ±0.2% |
| p1t | Ambient pressure (total) | ±0.2% |
| T25 | IPC outlet temperature | ±0.4% |
| p25 | IPC outlet pressure | ±0.25% |
| T3 | HPC outlet temperature | ±0.4% |
| p3 | HPC outlet pressure | ±0.25% |
| p43 | HPT outlet pressure | ±0.25% |
| p46 | IPT outlet pressure | ±0.25% |
| T5 | LPT outlet temperature | ±0.4% |
| N46 | LP shaft speed | ±0.1% |
| N44 | IP shaft speed | ±0.1% |
| N4 | HP shaft speed | ±0.1% |
| Wf | Fuel flow rate | ±2% |

The fuel flow was not used for model adaptation purpose but rather as input to run the model in the same condition as the real engine. It can be observed from Table 1 that all the sensors necessary to build the required adaptation scheme could be normally installed on a civil aircraft. Please note that in this work, diagnostics was performed with simulated flight data; analyses performed during pass-off tests on the ground would benefit from a larger sensor suite and possibly higher measurements accuracy, hence a different matching scheme could be selected.

The sensor set in Table 5 is denoted as Set1. In this work, the influence of sensors selection was studied by implementing the proposed method on other two possible sets of available measurements. Set2 includes the measurements in Set1 except N46 and p43, while Set3 lacks N44 and p46 measurements. Hence, the adaptation scheme has to discard two performance deviation factors for each of these two sensors sets. According to the methodology illustrated in [38] and the influence diagrams in Figure 5, $\Delta\eta_{HPT}$ and $\Delta\eta_{LPT}$ were removed from the Set2 scheme, while $\Delta\overline{W}_{IPC}$ and $\Delta\overline{W}_{LPT}$ were removed from the Set3 scheme.

### 4.1. Case 1—Production Scatter Effect

The results for the faulty engines with production scatter only are presented in Table 6. The percentage of correctly identified data points for each fault is shown for the 1600 simulated engines in the fleet (100 different engines with 16 levels of degradation severity). Although Normal and Very Low were considered two separate states in the BN nodes to increase the CPT estimation accuracy, the results are here combined for practical reasons, as any deviation within the production scatter can be considered negligible. The first thing observed was that the identification of HPC deterioration was less accurate, mainly due to the low influence that $\Delta\overline{W}_{HPC}$ had on the available measurements.

Both BN methods presented a very low percentage of false alarms, which is desirable. The BN alone was quite effective in identifying L level of IPC deterioration (M level was mostly misclassified as L), M level for HPT and IPT, and any level of deterioration for LPT.

**Table 6.** Correctly identified faults with production scatter.

|     |      | BN Only | Adaptive Model Only | Hybrid Method |
|-----|------|---------|---------------------|---------------|
| IPC | N/VL | 99%     | 86%                 | 99%           |
|     | L    | 72%     | 84%                 | 93%           |
|     | M    | 52%     | 88%                 | 98%           |
| HPC | N/VL | 99%     | 70%                 | 98%           |
|     | L    | 27%     | 73%                 | 71%           |
|     | M    | 30%     | 74%                 | 87%           |
| HPT | N/VL | 99%     | 75%                 | 99%           |
|     | L    | 63%     | 76%                 | 97%           |
|     | M    | 80%     | 72%                 | 92%           |
| IPT | N/VL | 99%     | 73%                 | 99%           |
|     | L    | 66%     | 69%                 | 91%           |
|     | M    | 80%     | 72%                 | 98%           |
| LPT | N/VL | 99%     | 94%                 | 100%          |
|     | L    | 75%     | 95%                 | 99%           |
|     | M    | 86%     | 94%                 | 99%           |
| BV  | N    | 99%     | -                   | 92%           |
|     | L    | 93%     | -                   | 94%           |
|     | M    | 94%     | -                   | 92%           |

Because of the correlation between some of the deviation parameters, up to 31% of the faults were misclassified by the adaptive model only. It is also interesting to note that by only using the model adaptation procedure, some extent of smearing effect was present, with up to 27% of the engines for different fault cases showing additional faults together with the correct one, in a component that was actually healthy.

The hybrid method considerably improved the diagnostics accuracy compared to the single BN and reduced the number of false alarms compared to the adaptive model alone. Further improvements in the BN classification could be achieved by e.g., increasing the number of states or the training space, but the purpose of this work was to assess the improvements given by the adaptive model layer with the same BN parameters and CPTs. An advantage compared to the adaptive model alone is that a fault in the bleed valve could not be isolated by the model adaptation approach, which simply detected a variation in performance characterized by deviations in several efficiency parameters. The combination of adaptive model and BN could however detect and identify a BV leakage although the accuracy was slightly lower than the single BN in case of engine production scatter.

Since the identification accuracy for some components was low, it is important to show also the detection and isolation accuracy, reported in Table 7. This refers to correct detection and isolation of the different faults irrespectively of the severity, i.e., it includes misclassifications between L and M. It is possible to see that the majority of misclassifications in Table 6 were due to a wrong fault severity attribution, which is not a major concern even though it would affect a downstream maintenance schedule analysis. Only for the HPC, a fairly high percentage of misdetections (i.e., faulty engines considered healthy) was observed with both BN methods.

**Table 7.** Detection and isolation accuracy with production scatter.

|       | BN Only | Adaptive Model Only | Hybrid Method |
|-------|---------|---------------------|---------------|
| IPC   | 91%     | 89%                 | 92%           |
| HPC   | 72%     | 85%                 | 77%           |
| HPT   | 92%     | 85%                 | 100%          |
| IPT   | 92%     | 83%                 | 99%           |
| LPT   | 91%     | 92%                 | 99%           |
| BV    | 97%     | -                   | 95%           |

*4.2. Case 2—Different Flight Conditions*

Table 8 shows the diagnostics results from different flights of the same engine, i.e., with no production scatter. Data from different flight conditions were corrected back to reference conditions following two methods. For the "BN only", measurements were corrected with respect to ambient temperature and pressure as suggested by Volponi [43]. The results for the "Adaptive model only" were obtained by running the adaptive model in flight conditions (*AD,flight**); while in the "Hybrid method", the full procedure illustrated previously in Figure 3 was applied to correct the data with respect to reference condition (*AD,ref**).

In this case, the BN alone did not perform well because, even though the effect of different ambient temperature and altitude was in part compensated by the parameters' correction, the scatter in measured data was still significant. It is expected that a larger and denser training space would be needed to achieve higher accuracy. The adaptive model showed a higher accuracy, always above 70%, but a notable smearing effect was still present, induced by sensor noise and mainly the uncertainty in calculated altitude and Mach number. The hybrid method outperformed both the single BN and the adaptive model, combining the benefits of both. This was true also for BV leakage identification.

As before, Table 9 presents the detection and isolation rate for the various faults. It is still evident from Table 9 how the BN structure used for the hybrid method, if employed alone, did not achieve satisfactory results. The hybrid method improved the accuracy even compared to the adaptive model.

**Table 8.** Correctly identified faults in different flight conditions.

|  |  | BN Only | Adaptive Model Only | Hybrid Method |
|---|---|---|---|---|
| IPC | N/VL | 95% | 76% | 98% |
|  | L | 50% | 78% | 85% |
|  | M | 53% | 80% | 93% |
| HPC | N/VL | 96% | 77% | 99% |
|  | L | 40% | 81% | 70% |
|  | M | 41% | 83% | 97% |
| HPT | N/VL | 96% | 74% | 99% |
|  | L | 45% | 77% | 99% |
|  | M | 47% | 80% | 97% |
| IPT | N/VL | 96% | 71% | 99% |
|  | L | 45% | 75% | 99% |
|  | M | 50% | 74% | 98% |
| LPT | N/VL | 96% | 90% | 99% |
|  | L | 49% | 91% | 100% |
|  | M | 51% | 90% | 100% |
| BV | N | 100% | - | 99% |
|  | L | 23% | - | 98% |
|  | M | 45% | - | 100% |

**Table 9.** Detection and isolation accuracy in different flight conditions.

|  | BN Only | Adaptive Model Only | Hybrid Method |
|---|---|---|---|
| IPC | 55% | 86% | 92% |
| HPC | 48% | 86% | 90% |
| HPT | 47% | 86% | 99% |
| IPT | 49% | 84% | 99% |
| LPT | 50% | 94% | 100% |
| BV | 37% | - | 99% |

*4.3. Effect of Sensors Set*

The Case 1 and Case 2 tests were repeated with reduced sensors sets. Table 10 presents the results for Case 1, i.e., in case of engine serial deviations, when Set2 and Set3 are employed for the BN only, the adaptive model (AM), and the combined method (AM+BN). When Set2 is used, the $\Delta\eta_{HPC}$ and $\Delta\eta_{HPT}$ rows in the influence matrix show 94% correlation, and the $\Delta\eta_{IPC}$ and $\Delta\eta_{IPT}$ rows have a correlation of 88%. Hence, in the BN alone, IPC, HPC, and IPT faults were mostly misclassified with Set2. A similar observation can be done for Set3, for which $\Delta\eta_{HPC}$ and $\Delta\eta_{HPT}$ present 94% correlation, which affects the classification in this case of HPC faults. Overall however, the accuracy for the BN alone was not much lower than with the full sensor Set1, with a significant decrement only in IPT and LPT. When the adaptive model alone was used with Set2 and Set3, the overall diagnostics accuracy was reduced, especially in the components whose deviation factor was removed from the matching scheme. A considerable degradation in accuracy compared to Set1 was also observed for the hybrid method with both sensor sets, since the errors in model prediction are propagated to the BN. However, the results were in most cases better than the BN alone, in particular for Set3, because although the matching scheme run with a reduced number of variables, the model outputs could still be used as soft sensors in the BN to provide the missing information.

**Table 10.** Identified faults with different sensors sets and production scatter.

|     |      | BN Set2 | BN Set3 | AM Set2 | AM Set3 | AM+BN Set2 | AM+BN Set3 |
|-----|------|---------|---------|---------|---------|------------|------------|
| IPC | N/VL | 99%     | 99%     | 84%     | 76%     | 95%        | 90%        |
|     | L    | 50%     | 44%     | 82%     | 11%     | 61%        | 57%        |
|     | M    | 54%     | 56%     | 86%     | 1.5%    | 81%        | 63%        |
| HPC | N/VL | 98%     | 98%     | 77%     | 76%     | 95%        | 90%        |
|     | L    | 44%     | 33%     | 79%     | 77%     | 36%        | 29%        |
|     | M    | 52%     | 46%     | 82%     | 81%     | 65%        | 53%        |
| HPT | N/VL | 99%     | 99%     | 98%     | 75%     | 95%        | 90%        |
|     | L    | 70%     | 69%     | 55%     | 76%     | 63%        | 72%        |
|     | M    | 84%     | 88%     | 48%     | 72%     | 81%        | 75%        |
| IPT | N/VL | 98%     | 98%     | 41%     | 61%     | 95%        | 90%        |
|     | L    | 50%     | 41%     | 45%     | 63%     | 68%        | 55%        |
|     | M    | 53%     | 68%     | 53%     | 65%     | 84%        | 71%        |
| LPT | N/VL | 99%     | 99%     | 98%     | 92%     | 95%        | 90%        |
|     | L    | 52%     | 49%     | 54%     | 14%     | 67%        | 63%        |
|     | M    | 62%     | 58%     | 48%     | 1%      | 70%        | 59%        |

The results for Case 2 with sensors Set2 and 3 are summarized in Table 11. The same considerations on sensors influence on detectable performance deviations can be made, and the same trends in accuracy loss were observed also for the engine in different flight conditions. It was evident how IPC and LPT deterioration were hard to detect with Set3. The hybrid method provided also in this case an overall higher accuracy than the BN alone, although the loss in accuracy from the results obtained with Set1 was higher.

**Table 11.** Identified faults with different sensors sets and flight conditions.

|     |      | BN Set2 | BN Set3 | AM Set2 | AM Set3 | AM+BN Set2 | AM+BN Set3 |
|-----|------|---------|---------|---------|---------|------------|------------|
| IPC | N/VL | 80%     | 79%     | 74%     | 75%     | 96%        | 80%        |
|     | L    | 32%     | 15%     | 76%     | 11%     | 83%        | 16%        |
|     | M    | 43%     | 38%     | 78%     | 2%      | 96%        | 31%        |
| HPC | N/VL | 80%     | 79%     | 78%     | 77%     | 95%        | 75%        |
|     | L    | 37%     | 23%     | 83%     | 81%     | 74%        | 29%        |
|     | M    | 36%     | 25%     | 85%     | 83%     | 91%        | 57%        |
| HPT | N/VL | 78%     | 78%     | 99%     | 74%     | 97%        | 84%        |
|     | L    | 37%     | 36%     | 49%     | 77%     | 89%        | 64%        |
|     | M    | 44%     | 43%     | 49%     | 80%     | 61%        | 64%        |
| IPT | N/VL | 78%     | 79%     | 42%     | 63%     | 94%        | 78%        |
|     | L    | 48%     | 46%     | 47%     | 71%     | 100%       | 66%        |
|     | M    | 41%     | 40%     | 57%     | 67%     | 99%        | 82%        |
| LPT | N/VL | 80%     | 80%     | 100%    | 93%     | 95%        | 90%        |
|     | L    | 37%     | 33%     | 50%     | 9%      | 96%        | 21%        |
|     | M    | 41%     | 39%     | 50%     | 0%      | 73%        | 33%        |

## 5. Discussion

The proposed hybrid method for fault diagnostics was proven to achieve satisfactory accuracy in the isolation and identification of various components deterioration in a fleet of engines, although severity identification for HPC deterioration presented up to 30% error. The combined adaptive model and BN approach showed an overall accuracy of 95% for detection and isolation, with 94% correct severity classifications for different flight conditions, and 92% for engines serial deviations, with particularly poor performance for the HPC. The results were higher in both cases than for the same BN applied directly on measured data without going through the first layer of model

adaptation. Since the model provided a correction with flight conditions and a first step of fault isolation and identification, the BN in the hybrid method ended up working on better data, from which the effect of measurement noise, production scatter, and ambient conditions was already removed. The advantage of the hybrid approach is also that a reduced dataset is necessary for estimating the CPT, since variations in flight conditions do not need to be accounted for, which reduces the initial computational burden. Several actions can of course be taken to improve the accuracy of the standalone BN, for example increasing the training space, pre-treating the data with more advanced correction methods, or increasing the number of nodes states. This work demonstrates however that if the BN layer is combined with a model adaptation layer, the final accuracy is higher even with limited training data and reduced network complexity.

The use of an adaptive performance model for estimating performance parameters deviations resulted in successful fault identification for most of the engines. However, a smearing effect was always observed due to the strong correlation between various performance parameters, and up to 39% of false alarms occurred. In addition, mechanical faults could not be isolated. Overall, the outcome of the model adaptation approach strongly depends on the available sensors; since the sensors-states matrix needs to be square in the identification problem, if a sensor is malfunctioning one state needs to be dropped, with a consequent reduction in diagnostics accuracy. For this reason, also the hybrid approach suffered with a limited sensor set whereas the BN alone was observed to be more robust to sensor unavailability. However, the overall accuracy of the hybrid method was still superior to the BN alone even with limited sensors.

Although the overall accuracy was higher, the hybrid method did not give better results in all cases, which leads to think that multiple solutions should run in parallel and a downstream information fusion system should use conflicting suggestions for making a final decision.

The proposed strategy can be a solution for both on-line and off-line monitoring and diagnostics, since the computational time for analyzing thousands of points is few seconds on a normal computer. However, building the BN requires a significant effort and experience, or a considerable amount of data for the network to learn by itself. A reliable and validated model may be essential to generate the required data since large amounts of real deterioration data are seldom available.

A limitation of this work is that only single component faults were considered, while in reality, several components are expected to degrade at the same time at different rates. Nonetheless, the combination of adaptive model and BN could be proven beneficial also in this case. As such, once a certain level of deterioration is estimated in one component, this can be incorporated into the model to generate a new baseline that filters out that component degradation, and the BN could identify any deviation from the new baseline caused for example by another part starting degrading or an abrupt fault in a valve or others. This has to be demonstrated however with a dedicated study.

## 6. Conclusions and Future Work

In this work, a combination of physics-based gas path analysis and a probabilistic approach for gas turbine diagnostics was presented and tested on a simulated fleet of engines. The fleet data were generated by means of an adaptive performance model and comprised of engine-to-engine serial deviations, different deterioration levels, and flight-to-flight variations. Performance model adaptation and a Bayesian network classifier were tested separately and in combination. The use of the adaptive model alone showed good results but a significant smearing effect was observed and the accuracy dropped if less sensors than performance deviation factors were used. The Bayesian network alone was more robust toward sensors availability but the results were not satisfactory, the detection and isolation rate being 89% for production scatter but only 47% for different flight conditions. The hybrid method was proven to be superior since it included the benefits of both and successful identification of components degradation was demonstrated, with 94% and 96% correct isolation rate in presence of engine-to-engine variations and flight-to-flight variations respectively.

For the future, a more advanced reasoning method will be developed to fuse information from the single approaches and the hybrid one, as well as information coming from cross-correlations of measurements in the fleet. Baseline adaptation over time to distinguish multiple components

degradation and other mechanical failures will also be included in future work. It is expected that an improved identification of simultaneous multiple faults and in presence of gradual deterioration will be achieved.

**Author Contributions:** Conceptualization, V.Z., A.D.F. and K.G.K.; methodology, V.Z., A.D.F. and M.S.; software, M.S. and K.G.K.; data processing and analysis, V.Z. and A.F.; writing—original draft preparation, V.Z.; writing—review and editing, A.D.F., M.S. and K.G.K.; supervision and project administration, K.G.K. All authors have read and agreed to the published version of the manuscript.

**Funding:** This research was funded by the Swedish Research Foundation under the national project DIAGNOSIS.

**Acknowledgments:** The authors would like to acknowledge the other project partners for their advices and support, and Xin Zhao for the support with the software development.

**Conflicts of Interest:** The authors declare no conflict of interest. The funders had no role in the design of the study; in the collection, analyses, or interpretation of data; in the writing of the manuscript, or in the decision to publish the results.

## Nomenclature

**Acronyms**

| | |
|---|---|
| AD | Adapted |
| BN | Bayesian network |
| BV | Bleed valve |
| CPT | Conditional probability table |
| DAG | Direct acyclic graph |
| GPA | Gas path analysis |
| HPC, HPT | High pressure compressor/turbine |
| IPC, IPT | Intermediate pressure compressor/turbine |
| ISA | International standard atmosphere |
| LPC, LPT | Low pressure compressor/turbine |

**Abbreviations**

| | |
|---|---|
| corr | Corrected |
| det | Deteriorated condition |
| flight | Flight condition |
| flight* | Flight faulty/deteriorated condition |
| ref | Reference condition |
| ref* | Reference faulty/deteriorated condition |

**Symbols**

| | |
|---|---|
| $c$ | Correlation factor |
| $H$ | Influence matrix |
| $L$ | Low |
| $M$ | Medium |
| $N$ | Normal |
| $N$ | Shaft speed |
| $P$ | Probability |
| $p$ | Pressure |
| $S$ | Fault severity |
| $T$ | Temperature |
| $VL$ | Very low |
| $\overline{W}$ | Flow capacity |
| $x$ | Performance deviation vector |
| $z$ | Measurements deviation vector |

**Greek letters**

| | |
|---|---|
| $\delta$ | Pressure coefficient |
| $\Delta$ | Deviation |

| η | Efficiency |
|---|---|
| θ | Temperature coefficient |

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
