# Peer review of "Probabilistic Model for Aero-Engines Fleet Condition Monitoring"

_aerospace, doi:10.3390/aerospace7060066_

Round 1

Reviewer 1 Report

The authors present a hybrid method for diagnostics of gas turbine engines. The topic is of interest in the aerospace/propulsion community, it is clearly written and presented.

Some minor comments.

1.Provide captions and units in the y-axis of Figs. 4 and 5. 

2.Table 5:shouldn't the noise level be +/- ?

Reviewer 2 Report

The manuscript presents a combination of physics-based gas path analysis and a probabilistic approach for gas turbine diagnostics. Authors have generated fleet data by means of an adaptive performance model and comprised of engine-to-engine serial deviations, different deterioration levels, and flight-to-flight variations. The simulation results confirm the effectiveness of the proposed approach. 

The topic of the paper fits the aims and scope of the journal. The paper is well-structured and the results are clearly described. In my view, the paper could be considered for publication in Aerospace after addressing the following minor rooms by the author: 

1- In the introduction section, the necessity of doing this piece of research should be more highlighted by utilizing and gap analysis. In other words, a justification should be added to the paper about the necessity of doing this piece of research by discussing the pros and cons of the proposed method. 

2- There are some minor grammatical errors in the paper (especially in the results analysis section). So, the authors are recommended to revise their work once again from the English language point of view. 
